# Stance Detection on Social Media with Background Knowledge

**Ang Li**[1,2], **Bin Liang**[1,2,4*], **Jingqian Zhao**[1,2], **Bowen Zhang**[5],
**Min Yang**[6]**, and Ruifeng Xu**[1,2,3*]

[1] Harbin Insitute of Technology, Shenzhen, China
[2] Guangdong Provincial Key Laboratory of Novel Security Intelligence Technologies
[3] Peng Cheng Laboratory, Shenzhen, China
[4] The Chinese University of Hong Kong, Hong Kong, China
[5] Shenzhen Technology University, Shenzhen, China
[6] SIAT, Chinese Academy of Sciences, Shenzhen, China
{angli,23S051022}@stu.hit.edu.cn, bin.liang@cuhk.edu.hk,
zhang_bo_wen@foxmail.com, min.yang@siat.ac.cn, xuruifeng@hit.edu.cn

## Abstract

Identifying users' stances regarding specific targets/topics is a significant route to learning public opinion from social media platforms. Most existing studies of stance detection strive to learn stance information about specific targets from the context, in order to determine the user's stance on the target. However, in real-world scenarios, we usually have a certain understanding of a target when we express our stance on it. In this paper, we investigate stance detection from a novel perspective, where the background knowledge of the targets is taken into account for better stance detection. To be specific, we categorize background knowledge into two categories: episodic knowledge and discourse knowledge, and propose a novel Knowledge-Augmented Stance Detection (KASD) framework. For episodic knowledge, we devise a heuristic retrieval algorithm based on the topic to retrieve the Wikipedia documents relevant to the sample. Further, we construct a prompt for ChatGPT to filter the Wikipedia documents to derive episodic knowledge. For discourse knowledge, we construct a prompt for ChatGPT to paraphrase the hashtags, references, etc., in the sample, thereby injecting discourse knowledge into the sample. Experimental results on four benchmark datasets demonstrate that our KASD achieves state-of-the-art performance in in-target and zero-shot stance detection.

## 1 Introduction

Stance detection has been essential in learning the public options from social media platforms, which aims to automatically identify the author's opinionated standpoint or attitude (e.g., *Favor*, *Against*,

---

* Corresponding Author

| Target: Brazil World Cup | Stance: Against |
|---|---|
| Text: ... In the future, the FIFA World Cup should only be held in countries within the top 15 GDP per capita. | |
| Target: Joe Biden | Stance: Favor |
| Text: The whole idea of POTUS ... #VoteBlue #Blue-Wave | |

Table 1: Two examples to show the episodic knowledge and the discourse knowledge. The first is from the VAST dataset and the second is from the P-stance dataset.

or *Neutral*, etc.) expressed in the content towards a specific target, topic, or proposition (Somasundaran and Wiebe, 2010; Augenstein et al., 2016; Stefanov et al., 2020). Existing work has achieved promising results on different types of stance detection tasks on text, based on conventional machine learning methods (Hasan and Ng, 2013; Mohammad et al., 2016; Ebrahimi et al., 2016) and deep learning methods (Sun et al., 2018; Zhang et al., 2020; Chen et al., 2021; Liang et al., 2022a).

However, identifying a stance on social media is still challenging because the background knowledge of targets is not included in the posts, and the content often comprises implicit information in a concise format. For this reason, it is necessary to integrate background knowledge into the stance learning of the target to enhance the ability of the model's stance detection by fully understanding the target. To better exploit the background knowledge of the target, we divide it into two types: *episodic knowledge* and *discourse knowledge*.

Episodic knowledge (Ma et al., 2019) refers to our understanding of a target, which is the basis for us to express our stance on a target. That is, when we express our stance on a topic, we usu-

ally have a certain understanding of it. Here, the episodic knowledge generally is not explicitly mentioned in the text. As the red part of the first example in Table 1 shows, The author's opposition to hosting the World Cup in Brazil can only be understood by knowing the background knowledge that Brazil's GDP per capita ranks lower than 15th. Previous research (Hanawa et al., 2019; He et al., 2022) has shown that Wikipedia is a good source of background knowledge. However, the limitation of input length within the language model makes it impossible to directly input lengthy Wikipedia articles.

In addition, in real-world social media platforms, users are accustomed to using nicknames to express certain targets. Therefore, we present discourse knowledge (Fang et al., 2021) to understand the expressions of acronyms, hashtags, slang, and references in social media texts. The blue part of the second example in Table 1 illustrates that "POTUS" in the text refers to the "President of the United States," and "#VoteBlue, #BlueWave" represents the Democratic Party with implied support for Joe Biden.

Incorporating background knowledge into stance detection on social media poses two major challenges. First, the required knowledge lacks ground truth labels. Second, the knowledge retrieval module necessitates an in-depth comprehension of expressions to retrieve relevant background knowledge rooted in semantics and incorporate the knowledge into the original text for making stance judgments. Typically, unsupervised algorithms lack these abilities. However, large language models (LLMs), such as ChatGPT[1], Bard[2], and LLaMA (Touvron et al., 2023), exhibit exceptional abilities in reading and generating text. They have been pre-trained on extensive Wikipedia data and hence possess an immense amount of knowledge. In this paper, we propose **K**nowledge-**A**ugmented **S**tance **D**etection Framework (KASD), leveraging ChatGPT to extract and inject the aforementioned two types of knowledge. We crawl Wikipedia pages related to each target and develop a heuristic retrieval algorithm based on topic modeling and an instruct-based filter to obtain episodic knowledge. To incorporate discourse knowledge, we employ instruct prompting to decipher acronyms, hashtags, slang, and references in the text and

rephrase the original text. We apply KASD to both fine-tuned model and large language model, and conduct experiments on four benchmark stance detection datasets. The results show that, on the fine-tuned model, KASD outperforms the baseline models which have additional designs or background knowledge incorporation. On the large language model, the results demonstrate that knowledge retrieval-augmented ChatGPT based on KASD can effectively improve the performance on stance detection. Additionally, we find that with KASD distilling the understanding and relevant background knowledge of the large language model, the fine-tuned model can achieve better results with significantly fewer parameters.

The main contributions of our work are summarized as follows:

1) We investigate stance detection from a novel perspective by exploring background knowledge for an adequate understanding of the targets. The background knowledge is divided into episodic knowledge and discourse knowledge for better learning of stance features.

2) We design the KASD framework, which leverages ChatGPT to heuristically retrieve episodic knowledge and incorporate discourse knowledge.

3) A series of experiments have demonstrated that our knowledge-augmentation framework can effectively improve the accuracy and generalization ability of the fine-tuned model and large language model on stance detection[3].

## 2 Related Work

**Incorporating Episodic Knowledge**
Current retrieval methods typically employ keyword-based filtering (Zhu et al., 2022b) or direct use of knowledge graphs (Liu et al., 2021) for knowledge retrieval. However, these retrieval methods necessitate that the required background knowledge overlaps with the text, which is not always the case and could result in poor retrieval effects. Conforti et al. (2022) introduced financial signals as background knowledge to improve the stance detection of the WTWT dataset. Zhu et al. (2022a) leveraged unannotated data with a variational auto-encoding architecture for detecting vaccine attitudes on social media. The knowledge incorporated in these works lacks generality.

**Incorporating Discourse Knowledge**

---

[1]https://openai.com/blog/chatgpt/
[2]https://bard.google.com/

[3]The source code of this paper is available at https://github.com/HITSZ-HLT/KA-Stance-Detection

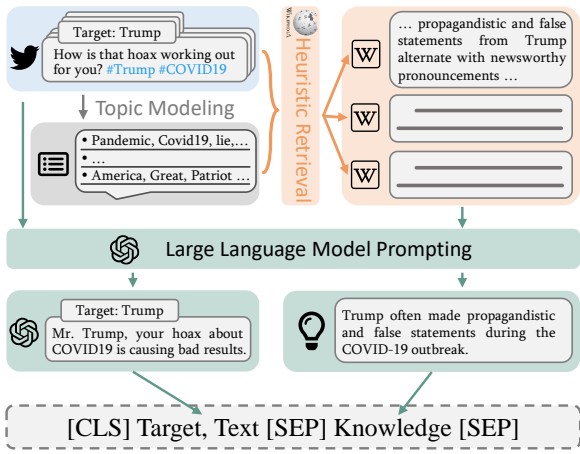

Figure 1: The architecture of our KASD framework.

Zhang et al. (2020) proposed a framework that extracts word-level semantic and emotional understanding to facilitate knowledge transfer across various targets. Ghosh et al. (2019) splited hashtags into individual words and employed substitute vocabulary to clarify these expressions. Zheng et al. (2022) proposed a prompt-based contrastive learning approach to stimulate knowledge for low-resource stance detection tasks. Xu et al. (2022) and Li and Yuan (2022) utilized data augmentation to enhance the model's comprehension of both text and target. Huang et al. (2023) introduced a background knowledge injection to modeling hashtags and targets. As experiment results reported in this paper, their approaches are suboptimal.

## 3 Methodology

Given $X = \{x_n, t_n\}_{n=1}^N$ as the labeled dataset, where $x$ denotes the input text and $t$ denotes the corresponding target, the goal of stance detection is to identify the stance label $y$ for the corresponding target $t$. $\{w_n^1, w_n^2, ..., w_n^M\} = x_n$ represents each word in the given text. As shown in Figure 1, we aim to retrieve multiple episodic knowledge $\{\mathcal{E}_n^i\}_{i=1}^K$ needed for the sample $x_n$, and inject the required discourse knowledge into $x_n$, resulting in $\mathcal{D}_n$. To achieve knowledge augmentation, we detect the stance of the sample using $\{t_n, \mathcal{D}_n, \mathcal{E}_n^1, ...\mathcal{E}_n^K\}$.

### 3.1 Episodic Knowledge Acquisition

For episodic knowledge, followed Zhu et al. (2022b); He et al. (2022), we conduct our knowledge base from Wikipedia. We retrieve the top 10 most relevant Wikipedia pages for each target using

the Wikipedia API[4]. Each Wikipedia page typically contains between 2,000 to 20,000 words, which exceeds the capacity of most encoding models. To differentiate among the various episodic knowledge, we segment each section of the Wikipedia page into separate documents. This segmentation allows us to group relevant information and assign an average length of approximately 400 words per document. Furthermore, this approach is readily extensible, with new targets easily added to the knowledge base using the same method.

### 3.2 Retrieval and Filtering

Existing method (Zhu et al., 2022b) typically begins with word-based retrieval, treating the episodic knowledge as the posterior $P(\mathcal{E}_i|x_n)$ of the $x_n$, and uses keywords in the text for retrieval. However, we argue that authors form their opinions on a target with a certain stance based on underlying topics and express these opinions accordingly. Therefore, the episodic knowledge behind these topics should be treated as the prior of the text $x_n$:

$$P(x_n|d_n) = \sum_k^M P(x_n|\mathcal{E}_i) \times P(\mathcal{E}_i|d_n) \quad (1)$$

where document $d_n$ denotes the combination of words.

### 3.2.1 Topic Modeling

For each episodic knowledge $\mathcal{E}_i$, we assume it corresponds to a topic $\mathcal{T}_i$ related to the sample $x_n$. To model this prior relationship, we establish a topic model for each target $t_n$ and use the fitted topics to retrieve the relevant episodic knowledge.

We set $T_n$ as the number of topics, which is a hyper-parameter that affects the effectiveness of the topic model. For the modeling process, we employ the Latent Dirichlet Allocation (LDA) algorithm, assuming the word distribution of topic $\mathcal{T}_i$ denoted by $P(\beta^l|\mathcal{T}_i)$, where each word in the vocabulary $V$ assigned a probability $\beta^l \in [0,1]^V$ of belonging to the topic $\mathcal{T}_i$, and assuming the topic distribution of a document $x_n$ denoted by $P(x_n|\mathcal{T})$ represents the probability of each word in the document belonging to each topic. To estimate these two distributions, we employ the online variational Bayes algorithm (Hoffman et al., 2010) implemented in sklearn[5].

---

[4] https://pypi.org/project/wikipedia/
[5] https://scikit-learn.org/stable/
modules/generated/sklearn.decomposition.
LatentDirichletAllocation.html

### 3.2.2 Heuristic Retrieval

After obtaining the word distribution probability $P(\beta^l|\mathcal{T}_i)$ for each topic and the distribution probability $P(x_n|\mathcal{T})$ for each document belonging to a topic, we propose a heuristic TF-IDF retrieval algorithm that combines the prior topics $\mathcal{T}_i$ with the original sample $x_n$ to match Wikipedia documents in our knowledge base. Initially, we establish a TF-IDF model for all Wikipedia documents in our knowledge base. Then, we design a retrieval method by combining the topic probability distribution, arriving at a topic-based retrieval score for each document:

$$S_{\text{topic}} = \sum_i^T (P(x_n|\mathcal{T}_i) \times \sum_l^L (\text{tf-idf}(\beta^l) \times P(\beta^l|\mathcal{T}_i)))$$
(2)

where $L$ represents the number of words contained in the prior topic $\mathcal{T}$. In our experiments, we find that by selecting the top 50 words based on their probability distribution, all topics satisfy $\sum_l^L P(\beta^l|\mathcal{T}) > 0.99$, indicating that most words relevant to each topic are covered. Besides retrieving based on a prior topic, we also consider the unique expression patterns of each sample. This retrieval score is based on a sample-specific calculation:

$$S_{\text{text}} = \sum_m^M (\text{tf-idf}(w_n^m) \times P(w_n^m))$$
(3)

where $P(w_n^m)$ represents the normalization coefficient that balances with $P(\beta^l|\mathcal{T}_i)$ and both have a value of $1/M$. The final retrieval score is the sum of these two parts without the need for any coefficient control, as normalization has already been performed:

$$\text{Similarity} = S_{\text{topic}} + S_{\text{text}}$$
(4)

The selection of relevant Wikipedia documents $\mathcal{W}_n$ is ultimately controlled by a threshold:

$$\mathcal{W}_n = \{\underset{\mathcal{W}_i}{\arg}(\text{Similarity} > \text{threshold})\}$$
(5)

From our observations of each dataset, we ultimately choose a threshold of 0.02, which filters out the majority of irrelevant Wikipedia documents. By utilizing our proposed retrieval method, both the synthesis of the sample's prior themes and the differentiated expressions are considered.

### 3.2.3 Large Language Model Filtering

After the heuristic retrieval process, the relevant Wikipedia documents $\mathcal{W}_n$ may contain redundant information which brings a negative effect on both the injection of background knowledge and the determination of the stance. However, dividing the Wikipedia documents and subjecting them to a refined retrieval process may cause a significant loss of contextual information. Therefore, we leverage ChatGPT as a filter for episodic knowledge. We build the prompt as:

> USER: Sentence: $x_n$. Target: $t_n$. Wikipedia Document: $\mathcal{W}_i$. If [Wikipedia Document] is not related to the given [Sentence] and the given [Target], output None. Otherwise, summarize the knowledge from the document which related to the given [Sentence] and the given [Target].

One sample may have multiple relevant Wikipedia documents, resulting in several prompts. We input each prompt into ChatGPT, and if the response is None, we consider the document irrelevant to the sample and discard it. If the response is a filtered knowledge, we concatenate them to obtain the filtered episodic knowledge $\mathcal{E}_n$. Here, ChatGPT is only allowed to extract knowledge from Wikipedia documents, thus preventing leakage of its stance labels. The ablation experiments conducted in Section 5.3 show that filtering can significantly enhance stance detection compared to unfiltered knowledge.

### 3.3 Discourse Knowledge Injection

To take advantage of the advanced contextual understanding capability and internal knowledge of ChatGPT to inject discourse knowledge, we design a prompt that allows ChatGPT to paraphrase the sample $x_n$, supplementing its acronyms, hashtags, slang, and references, and yielding the knowledge integrated sample $\mathcal{D}_n$:

> USER: Sentence: $x_n$. Please expand the abbreviations, slang, and hashtags in the [Sentence] into complete phrases and sentences to restate the text.

Our experiment has demonstrated that injecting discourse knowledge in this way is more effective and capable of boosting the generalization ability of fine-tuned models than merely pre-training or utilizing a substitution dictionary.

## 3.4 Knowledge-Augmented Stance Detection

We utilize KASD on both the fine-tuned model and the large language model.

### 3.4.1 Fine-tuned Model Stance Detection

To demonstrate the effectiveness of KASD in knowledge augmentation, we use a simple structure for the Fine-tuned Model. We input the sample which injecting discourse knowledge and concating filtered episodic knowledge into the BERT model for encoding.

$$\boldsymbol{h}_n = \text{BERT}([\text{CLS}]t_n, \mathcal{D}_n[\text{SEP}]\mathcal{E}_n[\text{SEP}]) \quad (6)$$

Then, the representation $\boldsymbol{h_n}$ is fed into a softmax classifier, and predicts the distribution of stance.

$$\boldsymbol{p}_n = \text{softmax}(\boldsymbol{W}\boldsymbol{h}_n + \boldsymbol{b}) \quad (7)$$

where $\boldsymbol{p}_n \in \mathbb{R}^{\boldsymbol{d_p}}$ is the predicted stance probability of the input instance $\boldsymbol{x}_n$, $\boldsymbol{d_p}$ is the dimensionality of stance labels. $\boldsymbol{W} \in \mathbb{R}^{\boldsymbol{d_p} \times \boldsymbol{d_m}}$ and $\boldsymbol{b} \in \mathbb{R}^{\boldsymbol{d_p}}$ are trainable parameters. The representation is fed into a single fully connected layer and softmax layer to predict the stance label $\hat{y} \in \{\text{favor}, \text{against}, \text{neutral}\}$, which is optimized by a cross-entropy loss:

$$\min_{\Theta} \mathcal{L} = -\sum_{i=1}^{N}\sum_{j=1}^{d_p} y_i^j \log \hat{y}_i^j + \lambda\|\Theta\|^2 \quad (8)$$

where $y_n$ is the ground-truth stance label distribution of instance $x_n$, $\hat{y}_n$ is the estimated distribution, $\Theta$ denotes all trainable parameters of the model, $\lambda$ represents the coefficient of $L_2$-regularization.

### 3.4.2 Large Language Model Stance Detection

Although large language models may internally contain the background knowledge to detect the stance of samples, we believe that explicit knowledge retrieval augmentation would substantially enhance the large language models' efficacy. Therefore, we apply our KASD framework to large language models as well. To better compare with the baseline, we use the same prompt as Zhang et al. (2023) and applied KASD for knowledge augmentation.

## 4 Experimental Setup

### 4.1 Datasets

We conduct experiments on four benchmark datasets in stance detection including SemEval-2016 Task6, P-stance, COVID-19-Stance, and Varied Stance Topics. The statistics is shown is Table 2 and Table 3.

| Dataset | Target | Favor | Against | Neutral |
|---------|--------|-------|---------|---------|
| | HC | 163 | 565 | 256 |
| Sem16 | FM | 268 | 511 | 170 |
| | LA | 167 | 544 | 222 |
| | Biden | 3217 | 4079 | - |
| P-Stance | Sanders | 3551 | 2774 | - |
| | Trump | 3663 | 4290 | - |
| | Fauci | 492 | 610 | 762 |
| Covid19 | Home | 615 | 250 | 325 |
| | Mask | 190 | 400 | 782 |
| | School | 693 | 668 | 346 |

Table 2: Statistics of SemEval-2016 Task6, P-stance and COVID-19-Stance datasets.

| | Train | Valid | Test |
|---|-------|-------|------|
| Examples | 13477 | 2062 | 3006 |
| Unique Comments | 1845 | 682 | 786 |
| Zero-shot Topics | 4003 | 383 | 600 |
| Few-shot | 638 | 114 | 159 |

Table 3: Statistics of the VAST dataset.

SemEval-2016 Task6 (Sem16) (Mohammad et al., 2016) consists of tweets containing six pre-defined targets. Following Huang et al. (2023) and Zhang et al. (2023), we conduct an experiment on the three targets: *Hillary Clinton* (HC), *Feminist Movement* (FM), *Legalization of Abortion* (LA), as these targets have a larger number of samples.

P-stance (Li et al., 2021) consists of tweets related to three politicians: *Joe Biden* (Biden), *Bernie Sanders* (Sanders) and *Donald Trump* (Trump). As noted in their paper, samples labeled as "None" have a low level of annotation consistency. Similar to prior research, we eliminate samples labeled as "None".

COVID-19-Stance (Covid19) (Glandt et al., 2021) consists of tweets containing four pre-defined targets: *Anthony Fauci* (Fauci), *stay-at-home orders* (Home), *wear a face mask* (Mask) and *keeping school closed* (School).

Varied Stance Topics (VAST) (Allaway and McKeown, 2020) is for zero/few-shot stance detection and comprises comments from The New York Times "Room for Debate" section on a large range of topics covering broad themes. It has about 6000 targets, far more than the other three datasets.

### 4.2 Implementation Details

For the fine-tuned model, we employ the RoBERTa (Liu et al., 2019) as the encoding module and a fully connected layer with batch normalization and LeakyReLU as the classifier, namely KASD-BERT. The models are trained using an AdamW optimizer with a batch size of 16 for a

| | Sem16(%) | | | | P-stance(%) | | | | COVID19(%) | | | | |
|---|---|---|---|---|---|---|---|---|---|---|---|---|---|
| | HC | FM | LA | Avg | Biden | Sanders | Trump | Avg | Fauci | Home | Mask | School | Avg |
| **Fine-tuned Model** | | | | | | | | | | | | | |
| RoBERTa | 55.97 | 68.19 | 67.60 | 63.92 | 84.29 | 79.56 | 82.70 | 82.18 | 77.44 | 79.46 | 80.89 | 72.84 | 77.66 |
| BERTweet | 62.31 | 64.20 | 64.14 | 63.55 | 82.90 | 79.00 | 84.41 | 82.10 | 82.91 | 80.90 | 77.98 | 78.77 | 80.14 |
| KPT | 71.30♯ | 63.30♯ | 63.50♯ | 66.03♯ | 80.40♯ | 77.10 | 80.20♯ | 79.23 | 84.37 | 81.34 | 84.27 | 76.71 | 81.67 |
| RoBERTa-Ghosh | 55.19 | 62.02 | 70.41 | 62.54 | 83.54 | 79.35 | 84.04 | 82.31 | 77.29 | 78.07 | 82.58 | 75.75 | 78.42 |
| BERTweet-Ghosh | 56.72 | 64.46 | 64.80 | 61.99 | 82.72 | 77.65 | 84.60 | 81.66 | 82.05 | 83.97 | 80.81 | 75.89 | 80.68 |
| KEprompt | 77.10♯ | 68.30♯ | 70.30♯ | 71.90♯ | 84.40♯ | - | 83.20♯ | - | - | - | - | - | - |
| WS-BERT-Dual | 75.26 | 66.02 | 70.42 | 70.57 | 83.50♭ | 79.00♭ | **85.80**♭ | 82.77♭ | 83.60♭ | 85.00♭ | **86.60**♭ | 82.20♭ | 84.35♭ |
| KASD-BERT | **77.60** | **70.38**⋆ | **72.29**⋆ | **73.42**⋆ | **85.66**⋆ | **80.39** | 85.35 | **83.80** | **87.49**⋆ | **87.97**⋆ | 86.20 | **83.03** | **86.17**⋆ |
| **Large Language Model** | | | | | | | | | | | | | |
| ChatGPT | 78.90† | 68.70† | 61.80† | 69.80† | 82.80† | **80.80**† | 85.70† | 83.10† | 77.48 | 72.02 | 69.58 | 57.95 | 69.26 |
| KASD-ChatGPT | **80.92** | **70.37** | **63.26** | **71.52** | **84.59** | 79.96 | 85.06 | **83.20** | **77.64** | **72.47** | **77.24** | **59.20** | **71.64** |

Table 4: In-target stance detection experiment results on Sem16, P-Stance and COVID19 dataset. The results with ♯ are retrieved from (Huang et al., 2023), ♭ from (He et al., 2022), † from (Zhang et al., 2023). Best scores are in bold. Results with ⋆ denote the significance tests of our KASD over the baseline models at p-value < 0.05. Since the results based on ChatGPT are the same each time, a significance test cannot be conducted.

maximum of 30 epochs with a warm-up ratio of 0.2. A learning rate of 1e-5 and a weight decay of 1e-3 are utilized. We report averaged scores of 5 runs to obtain statistically stable results. For the Large Language Model, we utilize the gpt-3.5-turbo-0301 version of ChatGPT and set the temperature to zero, ensuring replicable. We use the same prompt as the baselines and applied our framework for knowledge augmentation, namely KASD-ChatGPT.

### 4.3 Evaluation Metric

Following previous works (He et al., 2022), we adopt the macro-average of the F1-score as the evaluation metric. P-stance is a binary classification task, where each sample is labeled either as 'favor' or 'against'. Thus, we report $F_{avg} = (F_{favor} + F_{against})/2$. For Sem16, we follow the setup in Mohammad et al. (2016) and report $F_{avg} = (F_{favor} + F_{against})/2$. For COVID19 and VAST, we follow the setup in Glandt et al. (2021); Allaway and McKeown (2020) and report $F_{avg} = (F_{favor} + F_{against} + F_{None})/3$. In in-target stance detection, we select the one target to divide training, validation and test sets, consistent with other baselines. In zero-shot stance detection, for the SemEval16 dataset, following Huang et al. (2023), we select two targets as training and validation sets and the remaining one as a test set. For the P-Stance dataset, following Huang et al. (2023); Liang et al. (2022b), we select two targets as training and validation sets and the remaining one as a test set. (Which is the "DT, JB->BS", "DT, BS->JB," and "JB, BS->DT" described in dataset paper (Li et al., 2021)). For the VAST dataset, we use their original zero-shot dataset settings. We use standard train/validation/test splits for in-target and zero-shot stance detection across the four datasets.

### 4.4 Comparison Models

The fine-tuned model baselines include vanilla RoBERTa (Liu et al., 2019), domain pre-trained model: BERTweet (Nguyen et al., 2020), prompt based model: KPT (Shin et al., 2020), joint contrastive learning framework: JointCL (Liang et al., 2022b), incorporating discourse knowledge method (Ghosh et al., 2019): RoBERTa-Ghosh and BERTweet-Ghosh, incorporating ConceptGraph knowledge model: KEprompt (Huang et al., 2023), and incorporating Wikipedia knowledge model: TarBK-BERT (Zhu et al., 2022b) and WS-BERT (He et al., 2022). It should be noted that WS-BERT uses the COVID-Twitter-BERT model, which is a large-sized model, for the Covid19 dataset. Thus, on the Covid19 dataset, all BERT-based baselines, including our KASD-BERT, are compared using the large model. Apart from that, all other baselines and our KASD-BERT are utilized on the base model. For large language models, we compare KASD-ChatGPT with ChatGPT (Zhang et al., 2023), which utilizes few-shot chain-of-thought prompt for in-target stance detection and zero-shot prompt for zero-shot stance detection. Therefore, we guarantee that comparisons with all baselines are fair.

## 5 Experimental Results

We conduct experiments on two different methods of stance detection: in-target stance detection,

| | Sem16(%) | | | | P-stance(%) | | | | VAST(%) |
|---|---|---|---|---|---|---|---|---|---|
| | HC | FM | LA | Avg | Biden | Sanders | Trump | Avg | Avg |
| **Fine-tuned Model** | | | | | | | | | |
| RoBERTa | 43.45 | 40.38 | 38.79 | 40.87 | 76.29 | 72.07 | 67.56 | 71.97 | 73.18 |
| BERTweet | 44.82 | 21.97 | 31.91 | 32.90 | 73.13 | 68.22 | 67.66 | 69.67 | 71.10 |
| JointCL | 54.80♮ | 53.80♮ | 49.50♮ | 52.70♮ | - | - | - | - | 72.3♮ |
| RoBERTa-Ghosh | 44.78 | 41.33 | 29.21 | 38.44 | 76.28 | 70.57 | 66.81 | 71.22 | 73.07 |
| BERTweet-Ghosh | 44.51 | 36.21 | 34.43 | 38.38 | 73.18 | 68.39 | 66.36 | 69.31 | 71.90 |
| TarBK-BERT | 55.10♯ | 53.80♯ | 48.70♯ | 52.53♯ | 75.49 | 70.45 | 65.80 | 70.58 | 73.60♯ |
| WS-BERT-Dual | 53.65 | 47.06 | 42.61 | 47.77 | 77.91 | 71.63 | 69.24 | 72.93 | 75.30♭ |
| KASD-BERT | **64.78**★ | **57.13**★ | **51.63**★ | **57.85**★ | **79.04**★ | **75.09**★ | **70.84**★ | **74.99**★ | **76.82**★ |
| **Large Language Model** | | | | | | | | | |
| ChatGPT | 79.50† | 68.40† | 58.20† | 68.70† | 82.30† | 79.40† | 82.80† | 81.50† | 62.30† |
| KASD-ChatGPT | **80.32** | **70.41** | **62.71** | **71.15** | **83.60** | **79.66** | **84.31** | **82.52** | **67.03** |

Table 5: Zero-shot stance detection experiment results on Sem16, P-Stance and VAST dataset. The results with ♮ are retrieved from (Liang et al., 2022b), ♯ from (Zhu et al., 2022b), ♭ from (He et al., 2022), † from (Zhang et al., 2023). Best scores are in bold. Results with ★ denote the significance tests of our KASD over the baseline models at p-value < 0.05. Since the results based on ChatGPT are the same each time, a significance test cannot be conducted.

which is trained and tested on the same target, and zero-shot stance detection which performs stance detection on unseen targets based on the known targets.

## 5.1 In-Target Stance Detection

We perform experiments on Sem16, P-Stance, and COVID-19 for in-target stance detection. The experimental results are presented in Table 4. It shows that our proposed KASD framework outperforms all baseline methods in terms of both average results across all datasets and for most of the targets. It indicates that our KASD framework can improve the ability of the fine-tuned model to determine the stance through knowledge augmentation, without modifying the model itself. The results also indicate that knowledge retrieval-augmented ChatGPT using KASD can improve its performance in the In-Target Stance Detection task.

## 5.2 Zero-Shot Stance Detection

We conduct experiments on Sem16, P-Stance, and VAST for zero-shot stance detection. The experimental results, as shown in Table 5, indicate that our proposed KASD framework can significantly improve zero-shot performance in fine-tuned models(the results of p-value on most of the evaluation metrics are less than 0.05). It suggests that our knowledge-augmented method is more effective in zero-shot tasks for fine-tuned models which can enhance the model's understanding and generalization capabilities. The results based on ChatGPT show that our KASD framework can also largely improve the performance of ChatGPT on the Zero-

Shot Stance Detection task.

## 5.3 Ablation Study

To examine the impact of episodic knowledge and discourse knowledge, we provide two types of our proposed KASD in the ablation study:
(1) "**w/o** $\mathcal{E}$" denotes without the filtered episodic knowledge.
(2) "**w/o** $\mathcal{D}$" denotes without the injected discourse knowledge.

We conduct experiments on the P-Stance dataset for In-Target Stance Detection and the VAST dataset for Zero-Shot Stance Detection. The results are presented in Table 6. It indicates that for in-target stance detection on the P-Stance dataset, removing either episodic knowledge or discourse knowledge resulted in a significant decrease in performance across all targets. This suggests that for the P-Stance dataset, which contains a significant amount of hashtags and slang used in tweet posts, discourse knowledge, which enhances understanding of the samples, and episodic knowledge, which helps them judge difficult samples through background information, are both effective. While the VAST dataset consists mostly of standardized online debate texts, which usually exhibit standard conventions, discourse knowledge did not significantly improve the performance. However, since the VAST dataset contains a large number of targets, related episodic knowledge can help the model more effectively understand the targets, thereby enhancing the model's stance detection ability. Furthermore, Case Study A provided further evidence to support the above observations.

|  | P-stance(%) | | | | VAST(%) |
|---|---|---|---|---|---|
|  | Biden | Sanders | Trump | Avg | Avg |
| KASD-BERT | **85.66** | **80.39** | **85.35** | **83.80** | **76.82** |
| **w/o** $\mathcal{E}$ | 83.41 | 78.54 | 85.03 | 82.33 | 74.53 |
| **w/o** $\mathcal{D}$ | 83.69 | 79.01 | 84.19 | 82.29 | 76.44 |
| KASD-ChatGPT | 84.59 | **79.96** | 85.06 | 83.20 | 67.03 |
| **w/o** $\mathcal{E}$ | 82.59 | 78.10 | 81.69 | 80.69 | 65.22 |
| **w/o** $\mathcal{D}$ | 82.87 | 77.79 | 83.09 | 81.25 | 66.79 |

Table 6: Experimental results of ablation study in detecting in-target stance on the P-Stance dataset, and zero-shot stance on the VAST dataset.

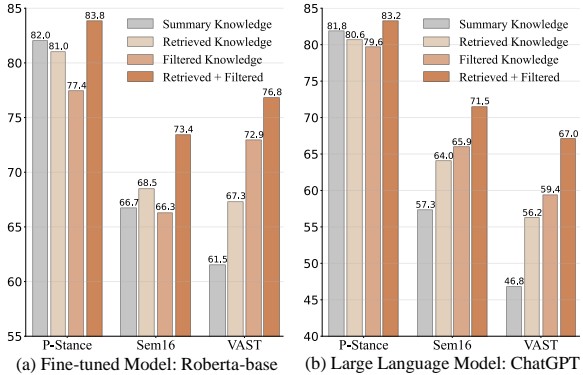

(a) Fine-tuned Model: Roberta-base    (b) Large Language Model: ChatGPT

Figure 2: Experimental results of three methods of retrieving background knowledge in detecting in-target stance on the P-Stance and Sem16 datasets, and zero-shot stance on the VAST dataset.

To validate the effectiveness of the heuristic retrieval algorithm and the filtering method designed in this paper, we conduct comparative experiments on three datasets, namely P-Stance, Sem16, and VAST. The experiments are conducted based on the fine-tuned models and the large language model, with four groups of comparisons:

- **Summary Knowledge:** Following the approach proposed by He et al. (2022), we only utilize the summary section from the Wikipedia page as knowledge.

- **Retrieved Knowledge:** We use the heuristic retrieval algorithm proposed in this paper to obtain the most similar Wikipedia document as knowledge.

- **Filtered Knowledge:** We use ChatGPT to directly extract episodic knowledge from the knowledge base without retrieval.

- **Retrieved + Filtered:** We use the framework proposed in this paper to retrieve and filter knowledge.

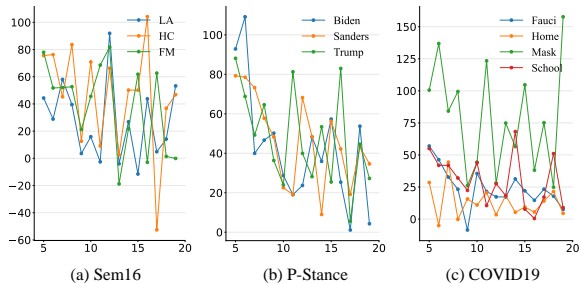

(a) Sem16    (b) P-Stance    (c) COVID19

Figure 3: The first-order differences of PPL across the number of topics $T_n$ ranging from 5 to 19 in Sem16, P-Stance, and Covid19 datasets.

The experiment results, as shown in Figure 2, indicate that redundancy may negatively impact the stance classification if Wikipedia documents are not effectively retrieved and filtered. Our retrieval and filtering structure in this paper can effectively obtain the necessary background knowledge for the samples, resulting in significant improvements.

### 5.4 Analysis of Topic Model

During the process of topic modeling, the number of topics $T_n$ acts as a hyper-parameter that affects the effectiveness of episodic knowledge. Perplexity (PPL) is a commonly used metric to measure the quality of language models. As the number of topics $T_n$ increases, the PPL of the topic model naturally increases. To address this, we use the first-order difference of PPL as an evaluation metric to evaluate the effectiveness of the Topic Model. Based on a preliminary understanding of the dataset, we limit $T_n$ between 5 and 19. The results are shown in Figure 3. For each target, we select the result with the minimum metric and set it as the value of $T_n$. Note that the VAST dataset features a small sample size for each target, rendering modeling an effective Topic Model difficult. Thus, the texts are utilized for the retrieval of episodic knowledge in the VAST dataset. From the results, we observe that for Sem16 and P-Stance, each target demonstrates strong topics. In contrast, for the Covid19 dataset, there are fewer topics, which is consistent with all targets in the Covid19 dataset that are nearly based on the same topic.

### 5.5 Human Evaluation

We randomly select 500 samples from the Sem16, P-Stance, Covid19, and VAST datasets and use human evaluation (with three evaluators who are not involved in this work) to measure the quality of the data generated by ChatGPT. Here, To assess the

|  | Generating episodic knowledge | Filtering redundant content | Generating discourse knowledge |
|---|---|---|---|
| Human Eval | 96.00% | 95.13% | 96.87% |

Table 7: Results of human evaluation on Sem16, P-Stance, Covid19, and VAST datasets.

|  | P-stance(%) | | | |
|---|---|---|---|---|
|  | Biden | Sanders | Trump | Avg |
| RoBERTa-large | 76.68 | 74.67 | 68.71 | 73.35 |
| BERTweet-large | 78.76 | 78.04 | 63.01 | 73.27 |
| ChatGPT | 82.30 | 79.40 | 82.80 | 81.50 |
| KASD-RoBERTa-large | **84.36** | **79.69** | **85.25** | **83.10** |

Table 8: Experimental results of RoBERTa-large, BERTweet-large, ChatGPT and KASD-RoBERTa-large detecting zero-shot stance on the P-Stance dataset.

quality of the generated episodic knowledge, we evaluate whether the filtered episodic knowledge is relevant to the respective sample and whether the filtered redundant content does not contain the required episodic knowledge. For the quality of the generated discourse knowledge, we evaluate the consistency of the generated discourse knowledge with the original content. The evaluators are asked to answer either "yes" or "no" to each of the three questions. Finally, we compute the mean proportion of "yes" responses from three evaluators for each question. A higher proportion indicates better data quality. The results are shown in Table 7.

The results show that ChatGPT is capable of generating high-quality background knowledge in the majority of cases (over 95%). This can be attributed to the fact that filtering redundant knowledge and generating discourse knowledge can be considered retrieval and generation tasks. Given that ChatGPT has been extensively trained on a substantial amount of similar data, consequently led to enhanced generation quality.

### 5.6 Fine-tuned Model vs Large Language Model

In this section, we compare the performance between the fine-tuned model and the large language model based on our KASD framework. Concerning in-target stance detection, Table 4 demonstrates that the effect of the RoBERTa-base model after knowledge augmentation is superior to ChatGPT model, which uses the few-shot chain of thought prompt. In zero-shot setup, Table 5 suggests that the RoBERTa-base model, after knowledge augmentation, performs better than the ChatGPT model on the VAST dataset. Additionally, Table 8 presents results for zero-shot stance detection on the P-Stance dataset, which is considered more challenging. It shows that the knowledge-augmented RoBERTa-large outperforms ChatGPT, with significantly fewer parameters the former employs, compared to the latter. These results imply that by distilling the large language model's understanding ability and background knowledge into a

smaller model through knowledge augmentation, the fine-tuned model can outperform the large language model with much fewer parameters(about 500~1000 times fewer).

ChatGPT's suboptimal performance can be attributed to its limited understanding of the stance detection task. The advantage of the fine-tuned model lies in its ability to better understand the task through task-specific supervised learning. However, the limited amount of training data hinders the development of general comprehension abilities. By utilizing KASD, we distill the understanding capabilities of large models and external knowledge in the forms of discourse knowledge and episodic knowledge, the performance of fine-tuned models can be effectively improved, thus surpassing large language models.

## 6  Conclusion

In this paper, we propose a Knowledge-Augmented Stance Detection (KASD) framework, providing heuristic retrieval and filtering of episodic knowledge and utilizing contextual information to inject discourse knowledge. We conduct experiments on in-target stance detection and zero-shot stance detection using four benchmark datasets. The experimental results demonstrate significant improvement in performance for both fine-tuned models and large language models utilizing KASD.

## Acknowledgments

We thank the anonymous reviewers for their valuable suggestions to improve the quality of this work. This work was partially supported by the National Natural Science Foundation of China (62006062, 62176076), Natural Science Foundation of Guang-Dong 2023A1515012922, Shenzhen Foundational Research Funding JCYJ20210324115614039 and JCYJ20220818102415032, Guangdong Provincial Key Laboratory of Novel Security Intelligence Technologies 2022B1212010005.

## Limitations

Our framework requires crawling relevant Wikipedia pages in advance to construct a knowledge base. We did not consider alternative sources, such as more timely news websites, for our knowledge base. Future research on exploiting this knowledge to explore timely stance detection methods is promising.

## Ethics Statement

The datasets used in this paper are sourced from open-access datasets. The VAST dataset provides complete text data in open access. In compliance with the privacy agreement of Twitter for academic usage, the Sem16, P-Stance, and COVID19 datasets were accessed using the official Twitter API[6] through the Tweet IDs to fetch complete text data. In these datasets, we touch on stance detection for some sensitive targets (e.g., belief, politics, etc.). Due to the noise of the training data and the imperfection of the models, unreasonable results may appear. We used the textual data obtained from Wikipedia and the ChatGPT service from OpenAI during the research process. We followed their term and policies.

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

## A   Case Study

We conduct a case study on Sem16, P-Stance, COVID19, and VAST datasets, and compared the results and knowledge with WS-BERT-Dual. The results are shown in Table 9, 10 and 11. We can observe that the background knowledge obtained based on ChatGPT has no label leakage and can help stance detection more effectively than other methods. We found that episodic knowledge can help the model solve difficult samples, while discourse knowledge can improve the generalization ability of the model and smooth out the differences between languages and expressions. This corresponds to the results of the ablation study.

| | Text (Target on: Climate Change is a Real Concern) | Stance |
|---|---|---|
| Original Sentence | Sea Level Rise above 6 meters - what does that mean? It means 20 ft above current heights. | favor |
| WS-BERT-Dual knowledge | Contemporary climate change includes both global warming and its impacts on Earth's weather patterns. There have been previous periods of climate change, but the current changes are distinctly more rapid and not due to natural causes. Instead, they are caused by the emission of greenhouse gases, mostly carbon dioxide ($CO_2$) and methane... | none |
| KASD discourse knowledge | Sea level rise above 6 meters means that it will be 20 feet above current heights. | favor |
| KASD episodic knowledge | humans face risks due to sea level rise, sea surface warming, and increased frequency and intensity of extreme weather events because of climate change. | |

| | Text (Target on: Hillary Clinton) | Stance |
|---|---|---|
| Original Sentence | @thehill : Women deserve a better candidate for the HIGH HONOR if first woman President: We ALL do! #WhyI'mNotVotingForHillary | against |
| WS-BERT-Dual knowledge | Hillary Diane Clinton (née Rodham; born October 26, 1947) is an American politician, diplomat, lawyer, writer, and public speaker who served as the 67th United States secretary of state from 2009 to 2013... | favor |
| KASD discourse knowledge | The Hill tweeted that women deserve a better candidate for the high honor of being the first woman President, and that everyone deserves a better candidate. This is the reason why the person is not voting for Hillary Clinton. | against |
| KASD episodic knowledge | None | |

| | Text (Target on: Legalization of Abortion) | Stance |
|---|---|---|
| Original Sentence | Antis don't care because the only people the laws are hurting are females and they don't care about things that don't effect men. | favor |
| WS-BERT-Dual knowledge | Abortion-rights movements, also referred to as pro-choice movements, advocate for legal access to induced abortion services including elective abortion. It is the argument against the anti-abortion movement. The Abortion rights movement seeks out to represent and support women who wish to terminate their pregnancy at any point... | favor |
| KASD discourse knowledge | People who oppose the legalization of abortion do not care because they only believe that females are negatively affected by these laws and they ignore issues that do not impact men. | against |
| KASD episodic knowledge | Margaret Sanger wrote: "No woman can call herself free until she can choose consciously whether she will or will not be a mother." From this perspective the right to abortion can be construed to be necessary in order for women to achieve equality with men whose freedom is not nearly so restricted by having children. | |

Table 9: Three examples from the Sem16 dataset.

| | Text (Target on: Donald Trump) | Stance |
|---|---|---|
| Original Sentence | #Trump planning to divert additional $7.2 billion in Pentagon funds for border wall @TeamPelosi @RepJerryNadler @RepSwalwell @SenSchumer | against |
| WS-BERT-Dual knowledge | Donald John Trump (born June 14, 1946) is an American politician, media personality, and businessman who served as the 45th president of the United States from 2017 to 2021... | favor |
| WS-BERT-Dual knowledge | President Trump is reportedly planning to use an additional $7.2 billion in funds originally allocated for the Pentagon's budget and put it towards constructing a wall at the southern border. | against |
| KASD episodic knowledge | In 2018, Trump refused to extend government funding unless Congress allocated $5.6 billion in funds for the border wall, resulting in the federal government partially shutting down for 35 days from December 2018 to January 2019, the longest U.S. government shutdown in history. | |

| | Text (Target on: Joe Biden) | Stance |
|---|---|---|
| Original Sentence | Now on OAN - Rudy is *interviewing* corrupt #Ukraine politicians - who were *sworn in* (in Ukraine) about Joe and Hunter #Biden ! | against |
| WS-BERT-Dual knowledge | "Joseph Robinette Biden Jr. (born November 20, 1942) is an American politician who is the 46th and current president of the United States. A member of the Democratic Party, he served as the 47th vice president from 2009 to 2017 under Barack Obama and represented Delaware in the United States Senate from 1973 to 2009... | favor |
| KASD discourse knowledge | Currently on One America News Network (OAN), Rudy Giuliani is conducting an interview with allegedly corrupt Ukrainian politicians who were inaugurated in Ukraine and are being questioned about the involvement of Joe and Hunter Biden. | against |
| KASD episodic knowledge | Since the early months of 2019, Biden and his father have been the subjects of unevidenced claims of corrupt activities in a Biden–Ukraine conspiracy theory pushed by then-U.S. President Donald Trump and his allies, concerning Hunter Biden's business dealings in Ukraine and Joe Biden's anti-corruption efforts there on behalf of the United States during the time he was vice president. | |

Table 10: Two examples from the P-Stance dataset.

| | Text (Target on: Gun Allowed College) | Stance |
|---|---|---|
| Original Sentence | A friend of mines father lost his leg fighting in Germany [Battle of the Bulge]. He never owned any firearms ever after the war......and scarcely spoke of it. Having to kill others with guns changes your life forever.... That's what war does. Do we really want to make universities feel like a war zone...... If you think having a classroom of armed students is going to make learning better.... Good Luck. | against |
| WS-BERT-Dual knowledge | A gun is a ranged weapon designed to use a shooting tube (gun barrel) to launch typically solid projectiles, but can also project pressurized liquid... | none |
| KASD discourse knowledge | A friend's father lost his leg in the Battle of the Bulge during the war in Germany. He was deeply affected by having to kill others and never owned any firearms after the war. War changes lives forever, and we should not want universities to resemble war zones. The idea of arming students in a classroom is unlikely to improve education, so it is not a good solution. | against |
| KASD episodic knowledge | Many do believe that permitting firearms in a classroom would lead to disruption in the learning processes of students but also diminish the overall safety of students. | |

| | Text (Target on: Olympics) | Stance |
|---|---|---|
| Original Sentence | Holding the games in Brazil is pure insanity. Zika and dengue fever are out of control. The Olympic committee expects swimmers to compete in an open sewer. Participants and fans will be risking their health, and the health of their families and even their future children, to attend these games, and also risk the health of their home countries upon return. All so that NBC & the olympic committee can make big bucks. It's not worth it. | against |
| WS-BERT-Dual knowledge | The modern Olympic Games or Olympics (French: Jeux olympiques) are the leading international sporting events featuring summer and winter sports competitions in which thousands of athletes from around the world participate in a variety of competitions.... | none |
| KASD discourse knowledge | The Hill tweeted that women deserve a better candidate for the high honor of being the first woman President, and that everyone deserves a better candidate. This is the reason why the person is not voting for Hillary Clinton. | against |
| KASD episodic knowledge | Some controversies during the Rio Olympics included the Zika virus epidemic and significant pollution in Guanabara Bay | |

Table 11: Two examples from the VAST dataset.