# OpenReview forum: "Stance Detection on Social Media with Background Knowledge"
_EMNLP/2023/Conference — EMNLP 2023 Main_

### Official Review · Reviewer_d5bz · 2023-08-05

**Soundness:** 3

**Excitement:**

3: Ambivalent: It has merits (e.g., it reports state-of-the-art results, the idea is nice), but there are key weaknesses (e.g., it describes incremental work), and it can significantly benefit from another round of revision. However, I won't object to accepting it if my co-reviewers champion it.

**Paper Topic And Main Contributions:**

This paper investigate stance detection from a perspective by exploring background knowledge for an understanding of the targets. The background knowledge is divided into episodic knowledge and discourse knowledge for learning of stance features. It design the KASD framework, which leverages ChatGPT to heuristically retrieve episodic knowledge and incorporate discourse knowledge.

**Questions For The Authors:**

（1）Why you use the topic model to retrieval the target page, not use the BERT embedding to look up the pages?
(2) Why the fine-tuned model perform better than the ChatGPT? How many parameters fine-tuned,10% ,30%,...,can exceed the ChatGPT?


**Reasons To Accept:**

(1) It is an interesting work to incorporate the two background knowledge to stance detection, KASD framework can conduct the task with a simple and effective way.
(2) The experimental result can well show the performance of KASD-based model, it can indeed improve the quality of the stance detection.

**Reasons To Reject:**

(1) The work only use a LLM, ChatGPT, you can also test it in LLaMA.
(2) In line 96, large language models(LLMs), such as ChatGPT, Bard, and LLaMA. It lacks the references about the three models

**Reproducibility:**

5: Could easily reproduce the results.

**Reviewer Confidence:**

5: Positive that my evaluation is correct. I read the paper very carefully and I am very familiar with related work.

---

> ### Author Rebuttal · Authors · 2023-08-29
>
> We truly appreciate the time and effort you've taken to review our work.
>
> >**Reject#1:**\
> >The work only use a LLM, ChatGPT, you can also test it in LLaMA.
>
> **Response of Reject#1:**\
> Thanks a lot for your valuable comments. Because the previous LLM-based work only used ChatGPT in this task, we primarily conducted experiments on ChatGPT. Based on your valuable suggestions, we also experimented with LLaMA-2-70b-chat for in-target stance detection and zero-shot stance detection. The comparison results are shown in Tables 1 and 2.
>
> |              | HC    | FM    | LA    | Avg   | Biden | Sanders | Trump | Avg   | Fauci | Home  | Mask  | School | Avg    |
> |:------------ |:-----:|:-----:|:-----:|:-----:|:-----:|:-------:|:-----:|:-----:|:-----:|:-----:|:-----:|:------:|:------:|
> | ChatGPT      |   78.90   |   68.70   |   61.80   |   69.80   |   82.40   | **80.80**   | **85.40** |   82.87   |   77.48   |   72.02   |   69.58   |   57.95    |   69.26    |
> | KASD-ChatGPT | **80.92** | **70.37** | **63.26** | **71.52** | **84.59** |   79.96     |   85.06   | **83.20** | **77.64** | **72.47** | **77.24** | **59.20**  | **71.64**  |
> | LLaMA2       |   72.91   |   65.04   |   49.71   |   62.55   |   78.57   | **71.64**   |   65.57   |   71.87   |   67.69   |   59.07   |   64.05   |   45.29    |   59.03    |
> | KASD-LLaMA2  | **77.89** | **67.29** | **52.00** | **65.73** | **79.59** |   71.32     | **67.89** | **72.93** | **71.38** | **67.33** | **67.10** | **52.89**  | **64.68**  |
>
> *Table 1: In-target stance detection experiment results on Sem16, P-Stance, and COVID-19 datasets based on ChatGPT and LLaMA2. Best scores are in bold.*
>
> |              | HC    | FM    | LA    | Avg   | Biden | Sanders | Trump | Avg   | VAST   |
> |:------------ |:-----:|:-----:|:-----:|:-----:|:-----:|:-------:|:-----:|:-----:|:------:|
> | ChatGPT      |   79.50   |   68.40   |   58.20   |   68.70   |   82.30   |   79.40     |   82.80   |   81.50   |   62.30    |
> | KASD-ChatGPT | **80.32** | **70.41** | **62.71** | **71.15** | **83.60** | **79.66**   | **84.31** | **82.52** | **67.03**  |
> | LLaMA2       |   74.24   | **67.19** |   55.92   |   65.78   | **76.24** |   73.67     |   64.47   |   71.46   |   39.85    |
> | KASD-LLaMA2  | **77.70** |   65.57   | **57.07** | **66.78** |   75.28   | **74.09**   | **69.27** | **72.88** | **43.42**  |
>
> *Table 2: Zero-shot stance detection experiment results on Sem16, P-Stance, and VAST datasets based on ChatGPT and LLaMA2. Best scores are in bold.*
>
> The results indicate that by using our KASD to enhance background knowledge, LLaMA2 could also achieve significant performance improvements in both in-target stance detection and zero-shot stance detection tasks. We will add the experiments and analysis to the final version.
>
> >**Reject#2:**\
> >In line 96, large language models(LLMs), such as ChatGPT, Bard, and LLaMA. It lacks the references about the three models
>
> **Response of Reject#2:**\
> Thank you for your valuable suggestions, we have added the corresponding references in the revision.
>
> >**Question#1:**\
> >Why you use the topic model to retrieval the target page, not use the BERT embedding to look up the pages?
>
> **Response of Questions#1:**\
> Thank you for your question. First, for the stance detection task, target-related information is crucial, and the text is distributed around several topics related to multiple targets. In this context, it is natural and effective to use a topic model to retrieve the knowledge pages. Secondly, we need knowledge pages to provide the model with additional information for stance judgment. However, the prerequisite for using BERT embedding to retrieve knowledge pages is that the knowledge and the original samples have the same semantics, which is not always the case. For example, as shown in Table 8 of Appendix A Case Study, the first sample is "Sea Level Rise above 6 meters—what does that mean? It means 20 feet above current heights, while the background knowledge needed is that human activities are causing rising sea levels and triggering extreme weather events and climate disasters. They are not strongly semantically related, but they describe the same topic. Therefore, our heuristic retrieval method based on the topic model can successfully retrieve the relevant information.
>
> >**Question#2:**\
> >Why the fine-tuned model perform better than the ChatGPT? How many parameters fine-tuned,10% ,30%,...,can exceed the ChatGPT?
>
> **Response of Questions#2:**\
> Thank you for your valuable comments about fine-tuning and your deep insight into this task. ChatGPT's suboptimal results can be attributed to its limited understanding of stance detection task. The advantage of the fine-tuned model lies in its ability to better understand the task through task-specific supervised learning. However, the limitation of the amount of training data prevents the development of general comprehension abilities. One promising method is distilling the understanding capabilities of large models and external knowledge into the forms of discourse knowledge and episodic knowledge so that the performance of fine-tuned models can be effectively improved, thus surpassing ChatGPT.\
> The question of how many parameters to fine-tune can exceed ChatGPT is influenced by various training strategies, such as which layer of network parameters to fine-tune or the use of parameter-efficient learning methods like prefix tuning, and exceeds the scope of this paper since we primarily focus on knowledge enhancement in stance detection tasks rather than complex model design and training strategies. Based on your suggestions, we conducted experiments on fine-tuning parameters on the 0th layer of RoBERTa (KASD-RoBERTa-large-layer-0) and the 11th layer of RoBERTa (KASD-RoBERTa-large-layer-11), two layers commonly used for study. The results are shown in Table 3.
>
> |                              |     Biden                   |      Sanders              |      Trump                    |       Avg                     |
> | :---                         |    :----:                   |    :----:                 |    :----:                     |    :----:                     |
> | ChatGPT                      | 82.30                       | 79.40                     |     82.80                     |     81.50                     |
> | KASD-RoBERTa-large           | 84.36                       | 79.69                     |     85.25                     |     83.10                     |
> | KASD-RoBERTa-large-layer-0   | 79.30                       | 74.26                     |     78.12                     |     77.23                     |
> | KASD-RoBERTa-large-layer-11  | 83.34                       | 79.12                     |     85.17                     |     82.54                    |
>
> *Table 3: Results of ChatGPT, KASD-RoBERTa-large, finetuning parameters on the 0th layer of RoBERTa (KASD-RoBERTa-large-layer-0) and the 11th layer of RoBERTa (KASD-RoBERTa-large-layer-11) detecting zero-shot stance on the P-Stance dataset.*
>
> From the results, we can observe that with knowledge-enhanced fine-tuned models, KASD-RoBERTa-large-layer-11 can outperform ChatGPT while only fine-tuning approximately 9% of its parameters. Thanks again for your comments; we will add the corresponding experiments and analysis to the final version.

---

### Official Review · Reviewer_sQss · 2023-08-06

**Soundness:** 4

**Excitement:**

4: Strong: This paper deepens the understanding of some phenomenon or lowers the barriers to an existing research direction.

**Paper Topic And Main Contributions:**

The paper introduces a new approach for identifying users' stances on specific topics from social media. While traditional stance detection relies on context alone, this method acknowledges that users often possess background knowledge about the subjects they discuss. The proposed technique, called Knowledge-Augmented Stance Detection (KASD) framework, leverages two types of background knowledge: episodic knowledge and discourse knowledge.
Episodic knowledge is obtained from relevant Wikipedia documents using a heuristic retrieval algorithm. ChatGPT is then prompted to filter and extract relevant information from these documents. For discourse knowledge, prompts are utilized to paraphrase hashtags, references, and other elements in the text, thereby integrating additional contextual knowledge.

The experimental results, based on four benchmark datasets, demonstrate that KASD outperforms existing methods in stance detection for both in-target and zero-shot scenarios.


**Reasons To Accept:**

- The topic of the paper is very interesting, hence worthy of investigation. It also aligns perfectly with the conference's theme.
- The paper is very well motivated and well written. The authors nicely motivate the work and position the paper in the literature.
- The experiment section is well-designed, the datasets and baseline methods are all appropriately chosen and well explained. The evaluation metric is nicely chosen and the experiment setup is detailed which make it reproducible.
- The numerical results shows significant improvement over the baseline methods which shows the effectiveness of the proposed framework.
- The authors also zoom in on various components of their framework to investigate their effectiveness which make their analysis more reliable and comprehensive.

**Reasons To Reject:**

Major:
- It seems like in Tables 4 and 5, some of the results are retrieved from other papers. Do the other papers have the same experiment setup as yours? Is this comparison fair and reliable? Please elaborate more on this in the paper.

minor:
- zero-shot learning is mentioned quite a few times in the text but is formally introduced in section 5 for the first time. I suggest introducing this definition earlier in the paper for better readability.
- typo line 353: argumentation

**Reproducibility:**

4: Could mostly reproduce the results, but there may be some variation because of sample variance or minor variations in their interpretation of the protocol or method.

**Reviewer Confidence:**

4: Quite sure. I tried to check the important points carefully. It's unlikely, though conceivable, that I missed something that should affect my ratings.

---

> ### Author Rebuttal · Authors · 2023-08-29
>
> We truly appreciate the time and effort you've taken to review our work.
>
> >**Reject#1:**\
> >It seems like in Tables 4 and 5, some of the results are retrieved from other papers. Do the other papers have the same experiment setup as yours? Is this comparison fair and reliable? Please elaborate more on this in the paper.
>
> **Response of Reject#1:**\
> Thank you for your insightful comments. Yes, we would like to clarify that our comparisons are fair and reliable. For the splitting of datasets and evaluation metrics, as mentioned in Section 4.3, we used the settings from the original dataset paper [1], [2], [3], and [4], which are also adopted by other baselines. The introduction to the model and training settings is in Section 4.4. For the configuration of fine-tuned models, we adopted the settings from [5] and [6], using the same model and training settings as them. As for the configuration of large language models, we followed [7], using the same model and prompts as them. We will provide detailed explanations of this aspect in the final version.
>
> [1]: [SemEval-2016 Task 6: Detecting Stance in Tweets](https://aclanthology.org/S16-1003) (Mohammad et al., SemEval 2016)\
> [2]: [P-Stance: A Large Dataset for Stance Detection in Political Domain](https://aclanthology.org/2021.findings-acl.208) (Li et al., ACL Findings 2021)\
> [3]: [Stance Detection in COVID-19 Tweets](https://aclanthology.org/2021.acl-long.127) (Glandt et al., ACL-IJCNLP 2021)\
> [4]: [Zero-Shot Stance Detection: A Dataset and Model using Generalized Topic Representations](https://aclanthology.org/2020.emnlp-main.717) (Allaway & McKeown, EMNLP 2020)\
> [5]: [Knowledge-enhanced Prompt-tuning for Stance Detection](https://dl.acm.org/doi/abs/10.1145/3588767) (Huang et al., TALLIP 2023)\
> [6]: [Infusing Knowledge from Wikipedia to Enhance Stance Detection](https://aclanthology.org/2022.wassa-1.7) (He et al., WASSA 2022)\
> [7]: [Investigating Chain-of-thought with ChatGPT for Stance Detection on Social Media](https://arxiv.org/abs/2304.03087) (Zhang et al.)
>
> >**Reject#2:**\
> >zero-shot learning is mentioned quite a few times in the text but is formally introduced in section 5 for the first time. I suggest introducing this definition earlier in the paper for better readability.
>
> **Response of Reject#2:**\
> Thank you for your careful reading and valuable comment. We will provide a comprehensive introduction to zero-shot stance detection within the Introduction section.
>
> >**Reject#3:**\
> >typo line 353: argumentation.
>
> **Response of Reject#3:**\
> Thank you for your comments. We have corrected 'argumentation' to 'augmentation' in the revision. We have thoroughly checked for spelling errors and will avoid them in future work.

---

### Official Review · Reviewer_ZhbM · 2023-08-09

**Soundness:** 4

**Excitement:**

4: Strong: This paper deepens the understanding of some phenomenon or lowers the barriers to an existing research direction.

**Paper Topic And Main Contributions:**

This paper explores background knowledge to assist in the stance detection task, which refers to information that is not explicitly expressed but is often informative for identifying the correct stance label. In particular, background knowledge includes episodic knowledge (prior knowledge that is not expressed in the text) and discourse knowledge (certain special expressions).

The main contribution of this work is that it addresses stance detection from an effective and new angle: introducing background knowledge.

**Questions For The Authors:**

QA: I imagine there might be error propagation when using ChatGPT to filter redundant knowledge and generate discourse knowledge. I am just curious whether the authors have done any analysis to measure the quality of data generated by ChatGPT?

**Reasons To Accept:**

This paper addresses stance detection from a novel angle by introducing background knowledge to aid in stance detection. The proposed approach has shown improvement over several baselines for both in-target stance detection and zero-shot stance detection. The work appears technically sound, and the contributions are clear.

**Reasons To Reject:**

There are low risks associated with accepting this work, possible reasons for rejection are minor, mostly related to the clarification of some questions that might make the work more convincing.
For instance, how does KASD compare when directly using ChatGPT to extract episodic knowledge?
For SemEval16, what is the performance for the remaining three targets, and how was the dataset split?
Moreover, the authors should provide more detail on how SemEval and PSTANCE are applied in zero-shot stance detection.

**Reproducibility:**

4: Could mostly reproduce the results, but there may be some variation because of sample variance or minor variations in their interpretation of the protocol or method.

**Reviewer Confidence:**

3: Pretty sure, but there's a chance I missed something. Although I have a good feel for this area in general, I did not carefully check the paper's details, e.g., the math, experimental design, or novelty.

**Typos Grammar Style And Presentation Improvements:**

The caption of Table 7 should be: “zero-shot stance” instead of "in-target stance"

---

> ### Author Rebuttal · Authors · 2023-08-29
>
> We truly appreciate the time and effort you've taken to review our work.
>
> >**Reject #1:**\
> >How does KASD compare when directly using ChatGPT to extract episodic knowledge?
>
> **Response of Reject #1:**\
> Thank you for your valuable and positive comments. In the preliminary experiment, we conducted a comparison of directly using ChatGPT to extract episodic knowledge. We found that our KASD performs better than directly using ChatGPT to extract episodic knowledge for both in-target stance detection and zero-shot stance detection. The results are shown in Table 1 and Table 2. We will add the experiments and analysis to the revision.
>
> |           | P-Stance | Sem16 | VAST  |
> |:--------- |:--------:|:-----:|:-----:|
> | KASD-BERT | 83.80    | 73.42 | 76.92 |
> | DUC-BERT  | 77.41    | 66.26 | 72.94 |
>
> *Table 1: The results of extracting episodic knowledge through our KASD and extracting episodic knowledge directly using ChatGPT (short for DUC) in detecting in-target stance on the P-Stance and Sem16 datasets and detecting zero-shot stance on the VAST dataset based on the BERT model*
>
> |              | P-Stance | Sem16 | VAST  |
> |:------------ |:--------:|:-----:|:-----:|
> | KASD-ChatGPT | 83.20    | 71.52 | 67.03 |
> | DUC-ChatGPT  | 79.58    | 65.93 | 59.37 |
>
> *Table 2: The results of extracting episodic knowledge through our KASD and extracting episodic knowledge directly using ChatGPT (short for DUC) in detecting in-target stance on the P-Stance and Sem16 datasets and detecting zero-shot stance on the VAST dataset based on ChatGPT.*
>
> >**Reject #2:**\
> >For SemEval16, what is the performance for the remaining three targets, and how was the dataset split?
>
> **Response of Reject #2:**
> 1) We apologize for not including this clarification in the paper due to page limitations. Following the baselines of KEprompt [1] and ChatGPT [2], we selected HC, FM, and LA as these targets have a larger number of samples. We also conducted experiments to evaluate KASD's performance on the remaining three targets on in-target stance detection and zero-shot stance detection. Because the Sem16 dataset [3] only provided zero-shot stance detection for the target "Donald Trump", it was only experimented with for zero-shot stance detection. The results shown in Table 3 and Table 4 demonstrate that our KASD also achieves state-of-the-art performance.
>
> |              |  A    |  CC   |
> |:------------ |:-----:|:-----:|
> | RoBERTa      | 65.40 | 43.08 |
> | BERTweet     | 68.12 | 41.30 |
> | WS-BERT-Dual | 71.57 | 57.31 |
> | KASD-BERT    | **72.32** | **61.47** |
> | ChatGPT      | 61.18 | 60.37  |
> | KASD-ChatGPT | **61.92** | **62.72** |
>
> *Table 3: The results of in-target stance detection on target "Atheism" (A) and "Climate Change is a Real Concern" (CC). Best scores are in bold.*
>
> |              |  A      |  CC     | DT      |
> |:------------ |:-------:|:-------:|:-------:|
> | RoBERTa      | 26.80   | 18.70   | 32.12   |
> | BERTweet     | 30.49   | 12.48   | 26.88   |
> | TarBK-BERT   | **56.20**   | 39.50   | 50.80   |
> | KASD-BERT    | 55.97   | **40.11**   | **54.74**   |
> | ChatGPT      | 60.94   | 53.03   | 62.30   |
> | KASD-ChatGPT | **63.95**  | **55.83**   | **64.23**   |
>
> *Table 4: The results of zero-shot stance detection on target "Atheism" (A), "Climate Change is a Real Concern" (CC) and Donald Trump (DT). Best scores are in bold.*
>
>
> 2) The SemEval16 dataset was split by its publishers [3] into training and test sets. For in-target stance detection, we divided the original training set into training and validation sets with an 8:2 ratio. For zero-shot stance detection, following [4], we selected 5 targets and divided them into training and validation sets with an 8:2 ratio, while the remaining targets were used as the test set. In the camera-ready version, we will include an introduction to dataset partitioning.
>
> [1]: [Knowledge-enhanced Prompt-tuning for Stance Detection](https://dl.acm.org/doi/abs/10.1145/3588767) (Huang et al., TALLIP 2023)\
> [2]: [Investigating Chain-of-thought with ChatGPT for Stance Detection on Social Media](https://arxiv.org/abs/2304.03087) (Zhang et al.)\
> [3]: [SemEval-2016 Task 6: Detecting Stance in Tweets](https://aclanthology.org/S16-1003) (Mohammad et al., SemEval 2016)\
> [4]: [JointCL: A Joint Contrastive Learning Framework for Zero-Shot Stance Detection](https://aclanthology.org/2022.acl-long.7) (Liang et al., ACL 2022)
>
> >**Reject #3:**\
> >The authors should provide more detail on how SemEval and PSTANCE are applied in zero-shot stance detection.
>
> **Response of Reject#3:**\
> We apologize for the lack of clarity on zero-shot stance detection due to page limitations. As described in lines 442-444, zero-shot stance detection performs stance detection on unseen targets based on known targets. For the SemEval16 dataset, following [1], we select five targets as training and validation sets and the remaining one as a test set. For the P-Stance dataset, following [1], we select two targets as training and validation sets and the remaining one as a test set. (Which is the "DT, JB->BS", "DT, BS->JB," and "JB, BS->DT" described in their paper.) This experimental setup is commonly employed in other zero-shot stance detection research.
> In the final version, we will include a detailed introduction to zero-shot stance detection.
>
> [1]: [JointCL: A Joint Contrastive Learning Framework for Zero-Shot Stance Detection](https://aclanthology.org/2022.acl-long.7) (Liang et al., ACL 2022)
>
> >**Question #A:**\
> >I imagine there might be error propagation when using ChatGPT to filter redundant knowledge and generate discourse knowledge. I am just curious whether the authors have done any analysis to measure the quality of data generated by ChatGPT?
>
> **Response of Questions#A:**\
> Thank you for your valuable comments about conducting an analysis to measure the quality of data generated by ChatGPT and for your deep insight into our work. We do agree that there might be error propagation when using ChatGPT to filter redundant knowledge and generate discourse knowledge. Therefore, in our preliminary experiment, we randomly selected a total of 500 samples from the Sem16, P-Stance, Covid19, and VAST datasets and incorporated human evaluation (with three evaluators who are not involved in this paper) to measure the quality of the data generated by ChatGPT. Here, To assess the quality of the generated episodic knowledge, we evaluated whether the filtered episodic knowledge is relevant to the respective sample and whether the filtered redundant content does not contain the required episodic knowledge. For the quality of the generated discourse knowledge, we evaluated the consistency of the generated discourse knowledge with the original content. The evaluators were asked to answer either "yes" or "no" to each of the three questions. Finally, we computed the mean proportion of "yes" responses from three evaluators for each question. A higher proportion indicates better data quality. The results are shown in Table 5.
>
> |             | Generating episodic knowledge| Filtering redundant content| Generating discourse knowledge |
> | :---        |    :----:                    |   :----:                   |   :----:                       |
> | Human Eval  | 96.00%                       | 95.13%                     |     96.87%                     |
>
> *Table 5: Results of human evaluation.*
>
> The results demonstrate that ChatGPT is capable of generating high-quality background knowledge in the majority of cases (with an average of over 95%). This can be attributed to the fact that filtering redundant knowledge and generating discourse knowledge can be considered retrieval and generation tasks. Given that ChatGPT has been extensively trained on a substantial amount of similar data, it is highly adept at performing these tasks, consequently leading to enhanced generation quality.
> Thanks again for your valuable comments; we will add the knowledge quality analysis to the final version.

---

### Meta-Review · Area_Chair_Yxj5 · 2023-09-11

**Recommendation:** 4

**Metareview:**

This paper proposes to incorporate background knowledge into stance detection models, which is retrieved from Wikipedia and by prompting ChatGPT. To increase reproducability, the authors have additionally added ablation results with the open-source LLM LLaMA-2 in their rebuttal. They perform experiments on several stance benchmarks and for different levels of supervision, and show empirically that their proposed setting works well. The reviewers agree that the paper is well executed, though the approach is relatively pedestrian, as it is in line with a lot of contemporary work on prompting ChatGPT.

---

### Decision · Program_Chairs · 2023-10-07

**Decision:**

Accept-Main

**Comment:**

This paper proposes to incorporate background knowledge into stance detection models, which is retrieved from Wikipedia and by prompting ChatGPT. To increase reproducability, the authors have additionally added ablation results with the open-source LLM LLaMA-2 in their rebuttal. They perform experiments on several stance benchmarks and for different levels of supervision, and show empirically that their proposed setting works well. The reviewers agree that the paper is well executed, though the approach is relatively pedestrian, as it is in line with a lot of contemporary work on prompting ChatGPT.